# Fighting Microbial Infections from *Escherichia coli* O157:H7: The Combined Use of Three Essential Oils of the *Cymbopogon* Genus and a Derivative of Esculentin-1a Peptide

**DOI:** 10.3390/antibiotics13010086

**Published:** 2024-01-16

**Authors:** Raffaella Scotti, Bruno Casciaro, Annarita Stringaro, Filippo Maggi, Marisa Colone, Roberta Gabbianelli

**Affiliations:** 1Biological Service, Italian National Institute of Health, 00161 Rome, Italy; raffaella.scotti@iss.it; 2Laboratory Affiliated to Istituto Pasteur Italia-Fondazione Cenci Bolognetti, Department of Biochemical Sciences, Sapienza University of Rome, 00185 Rome, Italy; bruno.casciaro@uniroma1.it; 3National Center for Drug Research and Evaluation, Italian National Institute of Health, 00161 Rome, Italy; annarita.stringaro@iss.it (A.S.); marisa.colone@iss.it (M.C.); 4Chemistry Interdisciplinary Project (ChIP) Research Center, School of Pharmacy, University of Camerino, 62032 Camerino, Italy; filippo.maggi@unicam.it

**Keywords:** *Escherichia coli* O157:H7, esculentin-1a, essential oil, *Cymbopogon* genus, antibacterial activity, biofilm, antimicrobial peptides, synergism

## Abstract

The absence of effective therapy against *Escherichia coli* O157:H7 infections has led to the need to develop new antimicrobial agents. As the use of synergistic combinations of natural antimicrobial compounds is growing as a new weapon in the fight against multidrug-resistant bacteria, here, we have tested new synergistic combinations of natural agents. Notably, we investigated a possible synergistic effect of combinations of essential oils and natural peptides to counteract the formation of biofilm. We chose three essential oils (i.e., *Cymbopogon citratus*, *C. flexuosus* and *C. martinii*) and one peptide already studied in our previous works. We determined the fractional inhibitory concentration (FIC) by analyzing the combination of the peptide derived from esculentin-1a, Esc(1–21), with the three essential oils. We also studied the effects of combinations by time–kill curves, scanning electron microscopy on biofilm and Sytox Green on cell membrane permeability. Finally, we analyzed the expression of different genes implicated in motility, biofilm formation and stress responses. The results showed a different pattern of gene expression in bacteria treated with the mixtures compared to those treated with the peptide or the single *C. citratus* essential oil. In conclusion, we demonstrated that the three essential oils used in combination with the peptide showed synergy against the *E. coli* O157:H7, proving attractive as an alternative strategy against *E. coli* pathogen infections.

## 1. Introduction

Conventional antibiotics have been playing a significant role in relieving infection by diseases, but antibiotic abuse and overuse has led to the emergence of resistance of microorganisms. This has become a serious health problem, reducing therapeutic options for treating bacterial infections [1]. In the last few years, as the need to develop new effective antimicrobial drugs has emerged, many natural products have been introduced into research projects to investigate their antimicrobial activities. Interest in natural products is due to their activities that allow them to replace some synthetic compounds with deleterious effects. Among the studied substances with the highest potential are plant essential oils (EOs) due to their longstanding use as antimicrobial agents [2]. EOs are liquid mixtures of volatile substances generally obtained from aromatic plants and used in food preservation. Among them, those containing compounds bearing phenolic groups were the most effective [3]. EOs are natural mixtures known to have activity against both Gram-negative and Gram-positive bacteria. The antimicrobial activity of EOs is due to their damage to the cell wall and cytoplasmic membrane of microorganisms and reduction in the proton motive force [4]. Other agents that attracted interest as alternative therapeutic drugs are antimicrobial peptides (AMPs), natural products commonly found in all living species [5].

Organisms of all species of life, when stimulated by natural or chemical compounds, have been shown to produce peptides as a defense against microorganisms that do not induce resistant mutant forms [6].

Antimicrobial peptides are low-molecular-weight peptides, whose main mechanism of action is the perturbation of the bacterial anionic plasma membrane and inactivation of intracellular targets. In particular, frog skin AMPs have attracted scientific interest in the last few decades. Glands of Anuran produce a considerable amount of peptides that kill a broad range of bacteria and other microorganisms damaging their plasma membrane [7,8,9]. Esculentins-1 are a class of peptides derived from frog skin consisting of 46 residues [10,11]. Recently, several studies have shown that Esc(1–21), a synthetic peptide corresponding to the first 20 amino acids of esculentin-1a, is effective against many bacteria and fungi [8,12,13,14,15], including *Escherichia coli* O157:H7 [9]. *E. coli* O157:H7 is a pathogen that causes food-borne infections and can lead to the potentially fatal hemolytic uremic syndrome. It has the ability to form a biofilm with the consequent increase in its survival. Biofilm is an aggregate of microorganisms living within an extracellular polymeric matrix that they produce. It is formed on biotic or abiotic surfaces and consists of different steps that lead to the formation of a three-dimensional architecture. Mature biofilms are highly resistant to antimicrobial agents and disinfectants [16]. According to the National Institutes of Health (NIH), it is estimated that up to 80% of human bacterial infections are associated with bacterial biofilms [17], which allow pathogens to evade host defenses [18]. This makes biofilms particularly important in clinical settings, and, consequently, there is a need to develop novel strategies that specifically target biofilms’ modes of growth. In the last few years, many studies have demonstrated the efficacy of EOs [19,20,21,22], as well as peptides [9,23,24], against bacterial infections due to biofilm.

Recently, combination therapy has become attractive, especially against bacterial infection. This new strategy is based on the combination of two different drugs and takes advantage of their possible synergistic action, where the effect is substantially greater than the sum of their parts. This is interesting, as not only could the doses of the use of the individual drugs potentially be reduced, with a consequent reduction in their toxicity, but the mixture could also be more effective than the single drugs [25]. The therapeutic use of the combination of natural compounds with synergistic activity has been known since ancient times [26]. Several studies have been undertaken to identify different combinations of drugs with increased efficacy and prevent the emergence of antibiotic resistance [27,28,29]. A large body of evidence highlighted the beneficial effects of using AMPs [23,30,31,32,33] and EOs [34,35,36,37,38,39] in conjunction with conventional antibiotics, showing a synergistic interaction.

This work aimed to investigate how the antibacterial activity of Esc(1–21) in *E. coli* O157:H7 varied when combined with three different EOs of the *Cymbopogon* genus, namely *C. citratus, C. flexuosus* and *C. martinii*. Notably, this is the first study to take into account the possible interaction between EOs and antimicrobial peptides in counteracting biofilm formation and bacterial growth.

## 2. Results and Discussion

### 2.1. Chemical Compositions of Essential Oils

We determined the chemical compositions of the three EOs via GC-MS analysis (Appendix A), while the main components exceeding 1.5% in each EO are illustrated in Table 1.

The main compounds of EOs derived from *C. flexuosus* and *C. citratus* were geranial (42.43 and 42.83%, respectively) and neral (33.22 and 33.84%, respectively), which are monoterpene aldehydes that combine to form the monoterpen citral [40]. The EO from *C. martinii* contained the terpenoid geraniol (81.41%) and its ester geranyl acetate (11.74%) as its main components. The antimicrobial activity of geraniol and citral is due to their functional groups and their affinity for microbial membranes [4,41].

### 2.2. Antibacterial and Antibiofilm Activity

The antimicrobial activity of EOs and the peptide on planktonic growth was analyzed using the microdilution assay, and the minimal inhibition concentration (MIC) values are reported in Table 2.

The MICs of EOs ranged from 0.2 to 0.8% in *E. coli* EDL933 and from 1.6 to 3.2% in *E. coli* K12 strains, while the MICs of Esc(1–21) were 8 and 4 µM in EDL933 and K12 strains, respectively. These values were slightly higher than those previously published in our works [9,22], probably due to the different batches of preparations of EOs and the peptide. Esc(1–21) had stronger antimicrobial activity than the reference antibiotic kanamycin. Furthermore, it was evident that EDL933 was less sensitive to the peptide than the non-pathogenic K12 strain; in contrast, EDL933 strain was less resistant to the action of the EOs than the K12 strain.

The checkerboard assay technique is the most common method used to determine in vitro drug interactions [42]. The fractional inhibitory concentration indexes (FICI), as reported in Table 3, referred to planktonic growth and biofilm. A combination of two compounds is defined as synergistic when the FICI is ≤ 0.5, antagonist when the FICI is ≥ 2 and additive when the FICI is > 0.5 but ≤ 1 [25,35,38]. Table 3 showed that all the mixtures had a synergistic interaction, with a notable reduction in the MICs of each component compared to when it was used alone.

In our study, to achieve an absence of planktonic growth in the 96-well plate, the concentration of the components used in the mixtures was much lower than that used with the individual compounds. Using concentrations ranging from 1/4 to 1/16 of MIC for EDL933 and from 1/8 to 1/32 of MIC for K12 strains, we obtained FIC indices much lower than 0.5, showing the high synergistic activity of the three combinations

*E. coli* O157:H7 can form biofilm, which acts as a reservoir of infection and is difficult to eradicate [43]. The antibiofilm activity of the three EOs and Esc(1–21) has been studied in our last two works [9,22]. In this study, the results obtained via crystal violet staining also showed the synergistic effect of the three combinations in the biofilm as the FIC indices [44] were all less than 0.5 (Table 3). To obtain these FICI values, we used concentrations ranging from 1/8 to 1/512 of MIC for EDL933 and from 1/16 to 1/256 of MIC for K12 strains.

The comparison of FIC values in biofilm versus planktonic growth also indicated that biofilm was more susceptible to mixtures than plankton. The greater sensitivity of the biofilm compared to plankton is important for fighting bacterial infections mainly due to the formation of the biofilm. Furthermore, the most effective combination appeared to be the mixture of Esc(1–21) and *C. martinii* EO, with both inhibiting planktonic cell growth and biofilm formation in EDL933.

### 2.3. Kill Curve Assay

To confirm the synergistic action of the mixtures, we used the kill curve assay. The time-dependent antibacterial activities of the three EOs, Esc(1–21) and their combinations were obtained by plotting a survival curve, using a concentration of 1/4 MIC. As shown in Figure 1A, bacteria of the EDL933 strain exposed to Esc(1–21) showed a growth curve almost identical to that of the control.

Among the individual EOs, the one that had the greatest effect on growth was *C. citratus*, with a recovery of vitality only 8 h after treatment, while the growth curve of bacteria obtained in the presence of *C. martinii* experienced recovery of vitality 6 h after the treatment. In addition, in all EOs/peptide combinations, the cell viability dropped to zero 2 h after treatment, failing to show minimum growth recovery even after 24 h. The different growth of bacteria observed when they were in the presence of a compound alone or in combination indicated a synergistic effect on the EDL933 strain. The bacteria after a 24 h treatment with the three mixes had an almost 2-log (CFU/mL)-fold decrease relative to growth in the presence of the most active compound [45]. The synergistic effect was not evident in the K12 strain (Figure 1B) with the *C. flexuosus* EO/Esc(1–21) treatment, where there was not a 2-log decrease between growth in the presence of the most active compound (*C. flexuosus*) compared to when used in the mixture. *C. flexuosus* was also the only EO that showed a different trend between the two strains. In fact, after a 24 h treatment, in EDL933, there was a level of survival comparable to both that of the control curve and that obtained in the presence of the peptide alone, while in strain K12, there was no survival. However, Figure 1B shows an overlap between the growth curves of untreated bacteria and those treated with the peptide.

In conclusion, the bactericidal activity of Esc(1–21) was significantly enhanced when used in combination with EOs, and the results obtained via the time–kill curves confirmed the synergistic effect of all combinations, for both the planktonic growth and the biofilm, on the EDL933 strain, as well as of two combinations on the K12 strain, except for the mixture *C. flexuosus* EO/Esc(1–21), which did not appear to be synergistic only on K12, as indicated by the FIC indexes.

### 2.4. Scanning Electron Microscopy (SEM)

To analyze any morphological alterations of bacterial cells, due to the growth in the presence of a concentration of 1/4 FIC of the mixtures, we examined the biofilm of the EDL933 and K12 strains via SEM.

The effectiveness of the three combinations is shown in Figure 2, where it is only possible to observe the formation of the biofilm in two untreated strains (A and E); in these cells, the biofilm appeared well structured, with no impaired rod-shaped cells.

In contrast, the treated bacteria showed only very few scattered cells, without an evident extracellular matrix. Furthermore, the treated bacterial cells showed a significant structural alteration with pore formation (B, C, D and F) and the alteration of cell length, indicating a decrease in membrane and biofilm integrity, without any substantial difference in the efficacy between the mixtures.

### 2.5. Inner Membrane Permeation

To verify the ability of the mixtures to alter the permeability of the cytoplasmic membrane of *E. coli* EDL933, we used the fluorescent probe Sytox Green, as previously reported [15]. The Sytox Green graphs obtained in the presence of both the single components and the mixtures are shown in Figure 3A and Figure 3B, respectively.

Based on its well-known membrane perturbation mechanism, the kinetics of the peptide were rapid (2–5 min) and dose-dependent. Graphs showed that the curves of the individual EOs (Figure 3A) had very different trends compared to the individual peptide, as the EOs more slowly permeabilize the membrane. In addition, among the EOs examined, *C. martinii* appeared to be the best at perturbing the membrane. This was in line with the results of the checkerboard assay, where the *C. martinii* EO/peptide mixture showed the lowest FICI on the EDL933 strain, indicating the strongest effect of this mix. The combinations were processed using scalar concentrations of the peptide and invariable concentrations of the EOs, with the latter corresponding to those that did not allow planktonic growth in combination with the peptide (FICI value). In particular, we used *C. flexuous* at 0.05% (*v*/*v*) (corresponding at 1/4 of its MIC), *C. citratus* and *C. martinii* at 0.1% (*v*/*v*) (corresponding to 1/4 and 1/8 of their MICs, respectively). The results showed that the permeability of the inner membrane reached its maximum after about 30 min, and there was a correlation between the mixture doses and the extent of the membrane disturbance. Furthermore, the graphs highlighted that all the combinations used at the corresponding FIC concentrations at which there was inhibition of bacterial growth had low membrane perturbation activity, which was not comparable to that obtained with the peptide or EOs used at the corresponding MICs.

The only exception was the *C. flexuosus* EO/peptide mixture, which had a similar effect to its EO when used alone. These data suggest that the EOs, by facilitating the penetration of the peptide into the cells, limit its perturbing action on the bacterial membrane.

### 2.6. Gene Expression Analysis

To understand the molecular mechanisms underlying the antibacterial activity of EO/peptide combinations, we examined the differential expression of several genes implicated in biofilm formation and responses to osmotic and oxidative stress in *E. coli* O157:H7. We compared the results obtained for bacteria treated with the mixtures with those resulting from treatment with the peptide or *C. citratus* alone. The latter EO was chosen as a reference for the individual EOs because, in preliminary studies conducted on a few genes, the three EOs had a similar effect on gene expression.

It is known that flagella are essential for biofilm formation because they participate in the first cell–surface interaction [43]. Thus, we focused on genes involved in the regulation and synthesis of the flagellar system (*flh*CD, *fli*C and *mot*AB) [46,47,48].

The results (Figure 4A) showed that the presence of the single peptide or *C. citratus* EO induced the flagellar regulators *flh*CD, and *fli*C and the motility genes *mot*A and *mot*B, but not the *fli*C gene, in the presence of *C. citratus* EO. Differently, the same genes, in the presence of the three mixtures, showed a level of expression similar to that of the control. Only the regulator *flh*C continued to be overexpressed, although to a much lesser extent, compared to the treatment with the peptide alone. Furthermore, only with the *C. martinii* EO/Esc(1–21) mixture were *mot*AB genes expressed more than those in the control. These results indicated that the flagellar system participated to a lesser extent in the inhibition of biofilm formation in the bacteria treated with the mixtures compared to the bacteria treated with the peptide alone [9].

Toxic nitric oxide (NO) is known to be a key mediator of biofilm dispersal in a variety of bacteria [49,50]. The nitrite reductase NirB can reduce the nitrite formed in the cytoplasmic space to NO [51]. Previously, we demonstrated [9] that the *nir*B gene is involved in the detachment of formed biofilm in the presence of the Esc(1–21). In this study, we observed that the *nir*B gene was upregulated in the presence of all mixtures, like with Esc(1–21) and *C. citratus* EO, with an increase in NO, suggesting that biofilm dispersal was still functional in bacteria treated with the combinations.

In the bacteria Poly (l-glutamic acid) (PGA) acts as an adhesin and is one of the three major exopolysaccharides providing shape and structural support to the biofilm. In the *pga* operon, the genes *pgaC* and *pga*D are required for the synthesis, while *pga*A and *pga*B are required for the export of the protein [52]. Figure 4 showed that *pga*C was always more expressed than the control, whilst *pga*ABD genes were expressed less in the presence of the mixtures compared to the treatment with the single peptide or EO. Therefore, although PGA could still be synthesized, it would not be efficiently exported out of the cell, making biofilm formation difficult.

In our previous work [9], we observed that Esc(1–21) treatment induced an increase in the expression of the flagellar system, and the continuous movement of flagella caused the inhibition of biofilm adhesion and development. Furthermore, the overexpression of *nir*B caused increased cell dispersion. In this study, biofilm inhibition, caused by the treatment of bacteria with EO/peptide combinations, was not affected by the flagellar system but was presumably due only to cell dispersion (overexpression of *nir*B). Among the regulatory genes of the flagellar system, only *flh*C continued to be overexpressed, although at a lower level than that observed through treatment with the peptide alone. Furthermore, the decrease in PGA, obtained by treating the bacteria with the mixtures, caused the instability of the biofilm, with the loss of its architecture. PGA is an essential protein for maintaining the structural integrity of the biofilm, and the downregulation of the *pga* genes hinders its formation [53,54].

Another protein that could contribute to the ability of bacteria to adhere to surfaces is OmpF [55], a porin protein regulated by osmolarity [56]. In bacteria treated with the peptide combined with *C. citratus* and *C. martinii*, the expression of *omp*F (Figure 4B) decreased compared to that of the bacteria treated with Esc(1–21) or not treated. These results suggested that these mixtures could also influence the ability to form biofilms by acting on the adhesion protein OmpF.

In this study, we also investigated the outer membrane proteins *osm*B [57], *osm*C [58,59] and *osm*E [60]. In the presence of the three mixtures, the expression level of the *osm* genes (Figure 4B) was lower than that obtained when the bacteria were treated with the peptide. This was similar to what was observed by Audrain et al., who showed the upregulation of *osm*B and *osm*C after treatment with the antimicrobial peptide ApoEdpL-W [61]. Furthermore, the *osm*B and *osm*E genes were less expressed compared to bacteria treated with *C. citratus* alone. Since *osm* genes are induced via an increase in osmotic pressure, their lack of induction during the treatment of the bacteria with the combinations suggested that there was no increase in osmotic pressure, as inferred in Esc(1–21) treatment [9]. The behavior of the *omp*F gene was in line with these results since it was not induced by EO/peptide combinations. It is known from the literature [56] that this gene is influenced by osmotic pressure. This was in line with the results obtained with Sytox Green, which highlighted how the EOs, combined with the peptide present in the mixtures, could facilitate the crossing of the membrane by Esc(1–21), perturbing it to a lesser extent and reducing osmotic shock.

One of the mechanisms of action of antibacterial compounds is the production of reactive oxygen species (ROS), with the consequent oxidative damage caused to DNA and proteins. As ROS are thought to contribute causatively to drug lethality [62,63,64], we investigated whether treatment with the mixtures could induce oxidative stress in bacteria with the production of ROS. The analyzed genes, *sod*C and *kat*E, encoding for the enzymes superoxide dismutase and catalase, respectively, are involved in the response to ROS and are induced by antimicrobial peptides [65,66]. The two mixes with *C. flexuosus* and *C. citratus* induced both *sod*C and *kat*E, while the combination with *C. martinii* only induced the *sod*C gene. Therefore, although to a lesser extent than the peptide alone, the combinations of EO and Esc(1–21) still induced oxidative stress, suggesting that the antimicrobial effect of the three compositions would be mediated by the induction of ROS cellular damage.

Taken together, the experiments show that the effect on gene expression of the three EOs combined with the peptide was essentially similar for all the genes analyzed. In only a few genes, the expression differed when the oily component of the mixture was *C. martinii*. Although the analysis of the genic expression level of some genes showed the greater or equal effectiveness of the individual components compared to those of the mixtures, the synergistic effect of the combinations would be more advantageous since reducing the doses of the individual components would reduce their toxicity. Furthermore, the results indicate that the inhibition mechanism of biofilm formation differed when we used Esc(1–21) alone or in combination with EOs. Indeed, in our previous work, we demonstrated that the peptide influences biofilm formation through the continuous production of flagella and the dispersion of the formed biofilm [9]. The results of this study instead demonstrate that the mixtures influenced the ability of the bacteria to form biofilms not only through the dispersion mechanism but also by destabilizing the biofilm during formation. Furthermore, the presence of EO/Esc(1–21) combinations induced oxidative stress but not osmotic stress.

## 3. Materials and Methods

### 3.1. Bacteria and Growth Condition

The *E. coli* EDL933 strain used in this study, which belonged to our laboratory collection [67], was representative of the enteropathogen *E. coli* O157:H7, and *E. coli* MG1655 was representative of the laboratory K12 strain.

Bacteria were grown in modified minimal medium M9 (modM9) at 28 °C [68].

### 3.2. Essential Oils and the Peptide

Commercial EOs lemongrass (*Cymbopogon citratus* and *Cymbopogon flexuosus*) and palmarosa (*Cymbopogon martinii*) were purchased from Naissure Trading (Neath, Dyfed, UK). The EOs’ sterility was tested on LB agar plates, and oils were stored at 4 °C in the dark. Stock solutions of EOs at 10% (*v*/*v*) were prepared by dissolving 50 µL of EO in 450 μL of dilution buffer (10% DMSO, 0.5% Tween 80 in PBS) before use.

Esc(1–21) was purchased from Bio-Fab Research (Rome, Italy) and assembled through stepwise solid-phase synthesis. Via reverse-phase high-performance liquid chromatography, we reached a purity of >95% of Esc(1–21). We verified the molecular mass through electron spray ionization mass spectrometry.

### 3.3. Gas Chromatography-Mass Spectrometry (GC-MS) Analysis

EOs was analyzed via gas chromatography-mass spectrometry (GC-MS), determining their chemical composition. We used an Agilent 8890 N chromatograph (GC) equipped with a single quadrupole 5977B mass spectrometer (Santa Clara, CA, USA) and an autosampler PAL RTC120 (CTC Analytics AG, Zwingen, Switzerland. The molecules were separated using an HP-5MS (30 m × 0.25 mm, 0.1 µm i.d.) capillary column purchased from Agilent, and helium (99.999%) was used as the mobile phase.

We reported, in Scotti R. et al. [9], the specific analytical conditions (injection, split mode, temperature program, scan mode), the identification and quantification of components.

### 3.4. MIC Assay

We determined the minimum inhibitory concentration (MIC) of the peptide Esc(1–21) and the three EOs, *C. citratus C. flexuosus*, and *C. martinii*, through the broth microdilution method [69], using the two-fold dilutions of each compound included the kanamycin reference antibiotic. We diluted an overnight inoculum in a total volume of 200 µL of modM9 to obtain the final concentration of 1 × 10^6^ CFU/mL, in the absence or presence of increasing concentrations of drugs. The concentration of EOs ranged from 0.1 to 6.4% (*v*/*v*), and the concentration of the peptide or kanamycin ranged from 1 to 32 µM. We inoculated the sample in triplicate in a 96-well polystyrene plate and incubated it at 28 °C, with constant agitation for 24 h. The MICs represented the lowest concentration of each compound at which no visible growth was observed.

### 3.5. Checkerboard Assay

To test the combined effect of EOs and the peptide, we used the fractional inhibitory concentration (FIC) index via the checkerboard method. The FIC index of two compounds, A and B, was calculated using the following formula: ΣFIC = FIC_A_ + FIC_B_, where FIC_A_ was obtained using the ratio between the value of the MICs of compound A alone and in combination; FIC_B_ was obtained in the same manner [42].

We inoculated a 96-well microplate with 1 × 10^6^ CFU/mL in a total volume of 200 µL per well. In the plate, each row (*x*-axis) contained the same diluted concentration of the compound A, while in the subsequent rows, the concentration of compound A was halved.

Similarly, each column (*y*-axis) in the plate contained the same diluted concentration of the compound B, while the concentrations in subsequent columns were halved. We incubated the microplate at 28 °C for 24 h. The lowest concentration of the compounds in wells with no visible growth was used to calculate the FIC index.

Two compounds are defined as synergistic when the FIC is ≤ 0.5, antagonistic when the FIC is ≥ 2 and additive when the FIC is > 0.5 but ≤ 1.

The checkerboard method was also used to estimate the Fractional Inhibition Concentration Index for biofilm cultures [44].

### 3.6. Biofilm Formation

A static biofilm formation assay was performed in 96-well polystyrene plates. We inoculated an overnight culture in a total volume of 200 µL of modM9 at a final concentration of 1 × 10^6^ CFU/mL and incubated at 28 °C for 24 h. We added to the cultures different EOs and peptide combinations based on checkboard assay. To quantify the total biofilm formation, the bacterial cell suspension was removed, and each well was washed three times with PBS to remove non-adherent cells and air dried for 1 h. We stained the biofilm with 0.1% Crystal Violet for 20 min and rinsed it three times with H_2_O. We measured the absorbance at 595 nm after the dye was dissolved in DMSO.

### 3.7. Time-Kill Curve

To study the bactericidal activity of mixtures as a function of time we used the time kill assay. Time–kill curves were performed, as previously reported [9], at a concentration of 1/4 MICs for each EO and peptide alone or in combination. An ON growth bacteria was added to modM9 at a final concentration of 1 × 10^6^ CFU/mL in a total volume of 200 µL in the presence or absence of drugs and incubated at 28 °C. A 100 µL aliquot was removed from each well, serially diluted and plated on LB agar after 0, 2, 4, 6, 8 and 24 h. Bacterial counts were detected after overnight incubation at 37 °C. The logarithm of CFU/mL at each time point was calculated. Each assay was replicated three times. Synergy was defined as a ≥2 log_10_ decrease in the colony counts caused by a combination compared to the most active single component at 24 h [45].

### 3.8. Permeation of the Bacterial Membrane

We used the fluorescent probe Sytox Green to verify the ability of the EO/peptide combinations to perturb the cytoplasmic membrane of *E. coli* EDL933, as previously reported [15].

Approximately 1 × 10^7^ CFU/mL, was incubated with 1 μM of Sytox Green in phosphate-buffered saline (PBS) for 5 min in the dark for the stabilization of the fluorescence signal. The cells were then treated with Esc(1–21) or EOs alone and their combinations. We monitored the fluorescence intensity, after the addition of the compounds, for 30 min using the microplate reader (Infinite M200, Tecan, Salzburg, Austria). On graphs, we plotted the recorded changes in fluorescence caused by the binding of the dye to intracellular DNA (λ_exc_ = 485 nm, λ_ems_ = 535 nm). The peptide concentrations ranged from 0.25 to 16 μM, and the EO concentrations ranged from 0.05 to 1.6%; for the combinations, the peptide was tested in scalar concentrations with invariable concentrations of the EOs, which corresponded to those that did not allow planktonic growth in combination with the peptide (FICI value).

### 3.9. RNA Isolation and Quantitative Real-Time RT-PCR

To isolate the RNA of the EDL933 strain used in the real-time RT-PCR experiments, we inoculated bacteria in 8 mL of modM9 at a final concentration of 1 × 10^6^ CFU/mL, in the presence or absence of mixes, used at a sub-lethal concentration of 1/4 FIC. For the RNA extracted from bacteria treated with Esc(1–21) or *C. citratus*, EO concentrations of 2 µM and 0.1% were used, respectively. Cultures were incubated at 28 °C for 24 h with 250 rpm agitation and stabilized with the RNA Protect Bacteria Reagent (Qiagen, Hilden, Germany). We used the Presto mini RNA Bacteria kit (Geneaid Biotech Ltd., New Taipei City, Taiwan) for the RNA extraction. We used DNaseI (Epicentre) for 20 min at 37 °C to eliminate the DNA contamination, and RNA was precipitated with 0.3 M sodium acetate and 0.7% isopropanol. We used PCR with specific primers to verify the absence of residual DNA. The quantity and integrity of RNA were determined using a UV-VIS one-drop micro-volume spectrophotometer (DeNovix-Resnova) at 260 nm. RT-PCR was used to investigate the transcription levels of different genes. These were genes involved in the dispersion of biofilm (*nir*B), flagellar synthesis (*flh*C, *flh*D, *fli*C) and motility (*mot*A and *mot*B), PGA synthesis and transport (*pga*A, *pga*B, *pag*C and *pga*D) and the oxidative (*sod*C and *kat*E) and osmotic stress responses (*omp*F, *osm*B, *osm*C and *osm*E). To perform RT-PCR, a SYBR Green kit (Luna Universal One-Step RT-qPCR, BioLabs) was used. The PCR cycling conditions and the standard curves were the same as those used in a previous study [9]. The forward and reverse primers used for gene amplification were designed using the Pel Primer software (Available online: https://perlprimer.sourceforge.net/download.html (accessed on 19 December 2023)) and are reported in Table 4 (those already used in the previous work were not reported) [9].

The efficiency of each new primer pair was determined using standard curves. R2 values or correlation coefficients < 0.95 were considered optimal correlations between values. We normalized the levels of target gene expression using the housekeeping gene 16*s* rRNA. We calculated changes in fold expression using the 2-ΔΔCT method [70]. Bacteria without treatments were used as a calibrator sample (control sample). We performed all RT-PCR experiments in at least triplicate.

### 3.10. Scanning Electron Microscopy (SEM)

Scanning electron microscopy (SEM) was used to assess the morphological effects of mixtures on biofilm formed by *E. coli* EDL933 and K12 strains. We used the combinations at sub-lethal concentrations of 1/4 FIC. Biofilms formed on 12 mm diameter glass coverslips were treated as previously described [22]. In brief, biofilms were fixed with 2.5% glutaraldehyde in sodium cacodylate buffer for 30 min and subsequently treated with 1% osmium tetroxide. These samples were dehydrated using a graded alcohol series and critical point-dried in CO_2_ (CPD 030 Blazers device, Bal-Tec, Blazers). The same samples were gold coated via sputtering (SCD 040 Blazers device, Bal-Tec) and observed using a scanning electron microscope FEI Quanta Inspect FEG, (FEI, Hillsboro, OR, USA).

### 3.11. Statistical Analysis

Experiments were performed in triplicate and repeated at least three times (*n* = 9). The results were presented as the averages ± the standard deviations, and statistically significant differences at *p* ≤ 0.05 (95%) and the confidence limit were determined via Student’s t-test using Microsoft Excel software (Microsoft 365).

## 4. Conclusions

The combination therapy seems to meet the need to find effective therapies in the clinical field and minimize the emergence of resistance. To the best of our knowledge, this is the first report of the synergistic activity of a natural peptide in combination with EOs. The present results show that *E. coli* O157:H7 and K12 strains, when grown in the presence of mixtures of the three *Cymbopogon* EOs and Esc(1–21), reduce their vitality and biofilm-forming ability. Furthermore, using these combinations, the effective concentration of the components can be significantly lower than when they are used alone, reducing their toxicity, decreasing the risk of monotherapy resistance, and allowing EOs to be used at organoleptically acceptable dosages. This study also indicates that EOs/peptide combinations act via different mechanisms than when the compounds are used alone, but further studies will be necessary to elucidate such a mechanism and understand the contributions of the main phytoconstituents of EOs derived from the *Cymbopogon* genus.

## Figures and Tables

**Figure 1 antibiotics-13-00086-f001:**
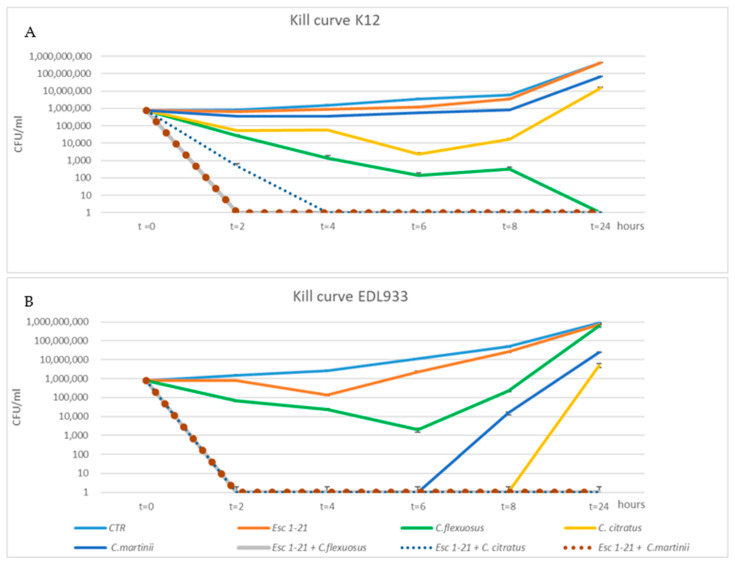
Time–kill curves of *C. citratus, C. flexuosus, C. martinii* and Esc(1–21) alone or in combination against EDL933 (**A**) and K12 (**B**) strains. Each component alone and in combination was used at ¼ of the respective MICs. The strains without any treatment were considered to be the control (CTR). The experiments were performed in triplicate. Data were expressed as means ± SDs.

**Figure 2 antibiotics-13-00086-f002:**
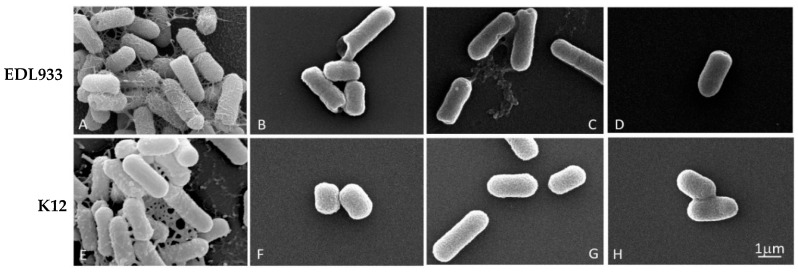
Scanning microscopy images of biofilm formed by *E. coli* strains in the absence or presence of EO/Esc(1–21) mixtures, used at a concentration of 1/4 FIC for 24 h. In detail, for the *C. flexuosus/*Esc(1–21) combination, we used 0.00156%/0.125 µM and 0.00312%/0.0156 µM for EDL933 and K12 strains, respectively; *C. citratus/*Esc(1–21) combinations of 0.0125%/0.0078 µM and 0.00625%/0.0625 µM were used for EDL933 and K12 strains, respectively; and *C. martinii*/Esc(1–21) combinations of 0.0125%/0.0039 µM and 0.0125%/0.0078 µM were used for EDL933 and K12 strains, respectively. (**A**,**E**) referred to the untreated EDL933 and K12 strains, respectively. The EOs used in combination with the peptide were *C. flexuosus* (**B**,**F**), *C. citratus* (**C**,**G**) and *C. martinii* (**D**,**H**).

**Figure 3 antibiotics-13-00086-f003:**
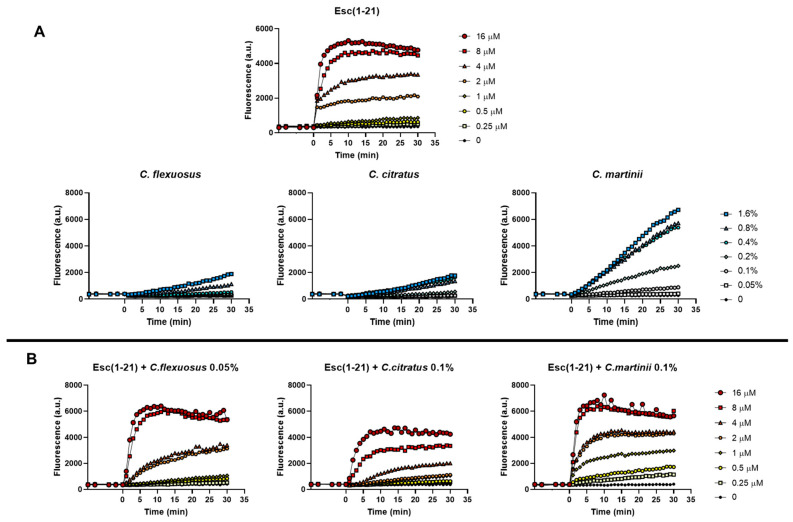
Effect of the presence of Esc(1–21), oils (**A**) and their combinations (**B**) on the membrane permeability of the EDL933 strain. The Sytox Green dye was used, and changes in fluorescence were monitored. Time 0 indicates the addition of the compounds. We performed three independent experiments and reported data from a single representative experiment.

**Figure 4 antibiotics-13-00086-f004:**
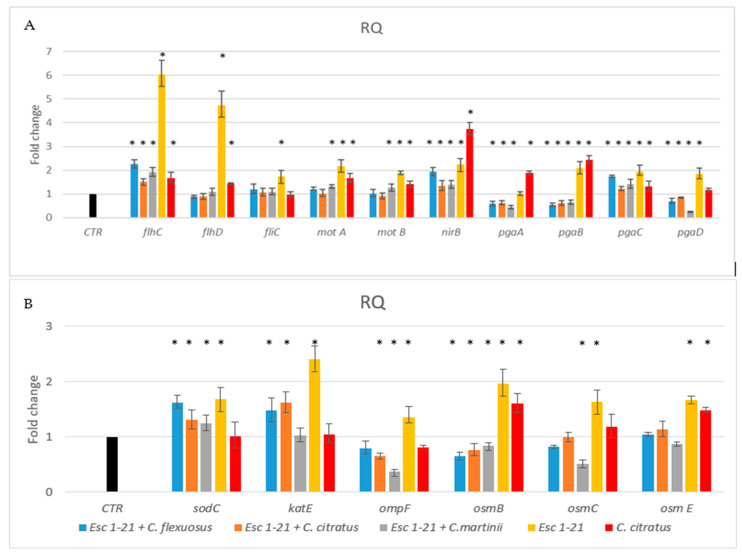
RT-PCR analysis of expression changes in genes related to flagellar system regulation, biofilm formation (**A**) and oxidative and osmotic stresses (**B**). Relative gene expressions (RQ) represent transcriptional levels after the exposure of *E. coli* EDL933 to singular components or mixtures versus the untreated control (CTR). The reference gene 16*s* rRNA was used to normalize the gene transcription level. Data are expressed as means ± SDs of three independent experiments performed in triplicate. * *p* ≤ 0.05.

**Table 1 antibiotics-13-00086-t001:** The main components present in the tested EOs at relative percentages higher than 1.5%.

EO	Main Components (%)
*Cymbopogon flexuosus*	geranial (42.43), neral (33.22), geraniol (5.17), geranyl acetate (3.29), (*E*)-caryophyllene (2.78), *iso*-geranial (2.52), camphene (1.81), (*z*)-*iso*-citral (1.55), γ-cadinene (1.42)
*Cymbopogon citratus*	geranial (42.83), neral (33.84), geraniol (4.63), geranyl acetate (3.49), *iso*-geranial (2.10), camphene (1.94), (*E*)-caryophyllene (1.83), 6-methyl-5-hepten-2-one (1.61), γ-cadinene (1.52)
*Cymbopogon martinii*	geraniol (81.41), geranyl acetate (11.74), linalool (2.32), (*E*)-caryophyllene (2.11)

**Table 2 antibiotics-13-00086-t002:** Minimal inhibitory concentrations (MICs). The results are expressed in µM for peptide and kanamycin and as percentages (*v*/*v*) for oils.

Strain	*C. citratus*	*C. flexuosus*	*C. martinii*	Esc 1–21	Kanamycin
EDL933	0.4%	0.2%	0.8%	8 µM	16 µM
K12	1.6%	3.2%	1.6%	4 µM	16 µM

**Table 3 antibiotics-13-00086-t003:** Fractional inhibitory concentration (FIC) indexes. FICIs refer to the planktonic growth (*) and biofilm (**). The values reported for each component of the mixture refer to the fraction of the MICs used.

**Strain**	**Esc 1–21+**	* **C. citratus** *	**FICI ***	**Esc 1–21 +**	* **C. flexuosus** *	**FICI ***	**Esc 1–21+**	* **C. martinii** *	**FICI ***
EDL933	1/16	1/4	0.312	1/8	1/4	0.375	1/16	1/8	0.187
K12	1/8	1/16	0.187	1/32	1/16	0.094	1/8	1/16	0.187
**Strain**	**Esc 1–21+**	** *C. citratus* **	**FICI ****	**Esc 1–21+**	** *C. flexuosus* **	**FICI ****	**Esc 1–21+**	** *C. martinii* **	**FICI ****
EDL933	1/256	1/8	0.129	1/16	1/32	0.094	1/512	1/16	0.064
K12	1/16	1/64	0.078	1/64	1/256	0.019	1/128	1/32	0.039

**Table 4 antibiotics-13-00086-t004:** Primers used in this study for reverse transcription-quantitative PCR.

Oligo Name	Sequence 5′-3′	
16*s*	For CATCCACAGAACTTTCCAGAG	Rev CCAACATTTCACAACACGAG
*mot*A	For GGGATTGGGTCGTTTATTGTC	Rev CATTGCTTTGGTGTATTTGGAG
*mot*B	For GCTGTAACCTTTCTCACCAC	Rev AATTGATAGAGTCCGATCCGA
*omp*F	For CTACCTATCGTAACTCCAACTTCT	Rev CCAAAGCCTTCGTATTCGTA
*osm*B	For GTTCTAACTGGTCTAAACGGG	Rev CCTAATGTACCCAACGTACTG
*osm*C	For CTTCACAATCGACCACATCC	Rev TTTAATGGACGGCAACTCAG
*osm*E	For TAACGCGACAAAGTAGGTTTC	Rev CGGCTTATGATCGTACCAAA
*pga*A	For GCTGAAGGTGTAATGGATAAAC	Rev AGGGACTGCGCATTGATTAC
*pga*B	For TGCGGTTAACTGACTTCACTTTAG	Rev ATAGCCATAATAGCGGTCCA
*pga*C	For GAATACAGCCTGACGACAATAT	Rev CAGCGATATGTGTCAATTCAATAT
*pga*D	For ATTATTACGACCCGACAATCAC	Rev GCCCAGACAATTAACACGAC

## Data Availability

Data are available on request.

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
