# Peer review of "Fighting Microbial Infections from Escherichia coli O157:H7: The Combined Use of Three Essential Oils of the Cymbopogon Genus and a Derivative of Esculentin-1a Peptide"

_antibiotics, 2024, doi:10.3390/antibiotics13010086_

Round 1
Reviewer 1 Report
Comments and Suggestions for Authors
The manuscript entitled „Fighting microbial infections from Escherichia coli O157:H7: the combined use of three essential oils of the Cymbopogon genus and a derivative of Esculentin-1a peptide” by Scotti et al. describes the antibacterial effects of a combination of essential oils and the antimicrobial peptide Esculentin-1a. The study is a follow-up on earlier reports describing the effects of peptide and essential oils alone on the growth of E. coli, now reporting synergistic effects when combined.
There are a few comments relating to the presentation of the data:
-There is a continuous change of font throughout the manuscript. This should have been already correctly formatted upon submission.
- Table 1: requires reformatting for better representation of the data.
- Table 3: It is highly recommended to show also the actual concentrations in µM for the peptide and % for oils (and dilution factor in brackets), otherwise it is hard for the reader to understand the results.
- Table 3: it would further be easier if the Table would be separated and has one header for planktonic growth and one for biofilm.
- Table 3: the synergistic effects are described for ¼ to 1/32 of the MIC´s in the text, but in Table 3 dilutions up to 1/256 are shown. The rationale for this should be described in the text.
- Figure 1: Again, the actual concentrations in addition to ¼ of MIC should be shown in the legend. The green line (Esc1-21 + C. flexuosus) is missing; therefore, the results are difficult to understand. If there is an overlay of lines in the graphics, a different representation of the data should be considered. Results with C. martinii should be also described.
- Second paragraph Page 4: Authors say that growth of EDL933 in the presence of Esc1-21 at this concentration is almost like the control. This also applies to K12, which is not mentioned in the text.
- Figure 2: show actual concentrations in addition to ¼ FIC in the legend.
- Figure 4: for a better understanding it is recommended to divide the figure into subfigures with groups of genes. Description of Y-axis is missing (e.g. fold change) and RQ should be defined in the figure legend.
- Materials and Methods: abbreviation modM9 needs to be defined.
Author Response
We thank the reviewer for his/her comment. Below we report the answers to his/her requests
Comment
There is a continuous change of font throughout the manuscript. This should have been already correctly formatted upon submission.
Answer:
We formatted all manuscript as requested
Comment
Table 1: requires reformatting for better representation of the data.
Answer:
The Table 1 was reformatted and now the data are better represented.
Comment
-Table 3: It is highly recommended to show also the actual concentrations in µM for the peptide and % for oils (and dilution factor in brackets), otherwise it is hard for the reader to understand the results.
-Table 3: it would further be easier if the Table would be separated and has one header for planktonic growth and one for biofilm.
-Table 3: the synergistic effects are described for ¼ to 1/32 of the MIC´s in the text, but in Table 3 dilutions up to 1/256 are shown. The rationale for this should be described in the text.
Answer:
We answered in order of simpleness of answers.
As requested, the Table 3 was separated into two panels, corresponding to planktonic growth (A) and biofilm (B).
In the text we better described the different ranges used for the EDL933 and K12 strains.
We deemed it unnecessary to further modify Table 3 for the following reasons:
- the inclusion of further data corresponding to the different concentrations of the individual components (expressed both in µM and in %) could complicate the Table without facilitate the understanding of results.
- since the MIC value of a substance could change following the use of different batches, expressing of the MIC in µM or as a % would give non-reproducible data compared to the use of ratios or multiples of the MIC, which instead remain invariable.
- it should be noted that Referee 2 did not request any changes of Table 3
- in the literature several authors used, in the various assays related to FIC values, ratios or multiples of MIC rather than the concentrations corresponding to the MIC. We have listed some of these works below
White RL, Burgess DS, Manduru M, Bosso JA. Comparison of three different in vitro methods of detecting synergy: time-kill, checkerboard, and E test. Antimicrob Agents Chemother. 1996 Aug;40(8):1914-8. doi: 10.1128/AAC.40.8.1914. PMID: 8843303; PMCID: PMC163439.
Petersen, P.J.; Labthavikul, P.; Jones, C.H.; Bradford, P.A. In vitro antibacterial activities of tigecycline in combination with other antimicrobial agents determined by chequerboard and time-kill kinetic analysis. J. Antimicrob. Chemother. 2006, 57(3):573-6. doi: 10.1093/jac/dki477. Epub 2006 Jan 23.
Kyaw BM, Arora S, Lim CS. Bactericidal antibiotic-phytochemical combinations against methicillin resistant Staphylococcus aureus. Braz J Microbiol. 2012 Jul;43(3):938-45. doi: 10.1590/S1517-838220120003000013. Epub 2012 Jun 1. PMID: 24031910; PMCID: PMC3768864.
Comment
Figure 1: Again, the actual concentrations in addition to ¼ of MIC should be shown in the legend. The green line (Esc1-21 + C. flexuosus) is missing; therefore, the results are difficult to understand. If there is an overlay of lines in the graphics, a different representation of the data should be considered. Results with C. martinii should be also described.
Answer:
We used 16 different concentrations for both strains (EDL933 and K12) and their inclusion in the capture or in the figure would complicate the reading. As correctly suggested by the reviewer, we have changed the color/symbol of the curves in the graphs to better highlight line overlaps. The results relating to C. martinii are now described in the text.
Comment
Second paragraph Page 4: Authors say that growth of EDL933 in the presence of Esc1-21 at this concentration is almost like the control. This also applies to K12, which is not mentioned in the text.
Answer:
We better explained in the text the trend of curves also of the K12 strain
Comment
Figure 2: show actual concentrations in addition to ¼ FIC in the legend.
Answer:
In this case, having fewer values to include in the legend, we chose to show the actual concentrations used, in addition to ¼ FIC, as suggested by the reviewer.
Comment
Figure 4: for a better understanding it is recommended to divide the figure into subfigures with groups of genes. Description of Y-axis is missing (e.g. fold change) and RQ should be defined in the figure legend.
Answer:
As suggested by reviewer, we divided the Figure 4 into two panels, A and B, separating the examined genes related to biofilm from those related to stress responses. Furthermore, we inserted the title of Y-axis in the graphs and RQ definition in the legend
Comment
Materials and Methods: abbreviation modM9 needs to be defined.
Answer:
The abbreviation modM9 was defined in Materials and Methods section
Reviewer 2 Report
Comments and Suggestions for Authors
The manuscript presents solution to one of the major health issues, the AMR. its is well written and will be quite impressive to readers.
The introduction part is quite lengthy and may be reduced deleting the unnecessary details.
the authors have enlisted the phytoconstituents of different plants but no attempts have been made to correlate them with the results.
Author Response
We thank the reviewer for this positive comment on the manuscript. Below we report the answers to his/her requests
Comment
The introduction part is quite lengthy and may be reduced deleting the unnecessary details.
Answer:
We realize that the introduction is quite lengthy but by eliminating some parts you may not fully understand the purpose and importance of the work, therefore we didn’t modify the Introduction.
Comment
the authors have enlisted the phytoconstituents of different plants but no attempts have been made to correlate them with the results.
Answer:
In this work, we have not considered this aspect as essential oils are very complex mixtures of phytoconstituents. The real contribution of the individual components will require further studies in which the corresponding chemical compounds will be used. This will be the aim of an already designed research project. In the Conclusions section we added a sentence that refers to this purpose.